# Interventions for subjective cognitive decline: systematic review and meta-analysis

Rohan Bhome,[1] Alex J Berry,[2] Jonathan D Huntley,[1] Robert J Howard[1]

[1]Division of Psychiatry, University College London, London, UK
[2]Camden and Islington NHS Foundation Trust, London, UK

**Correspondence to**
Dr Rohan Bhome;
rohan.bhome@ucl.ac.uk

## ABSTRACT

**Objectives** This review provides a broad overview of the effectiveness of interventions for subjective cognitive decline (SCD) in improving psychological well-being, metacognition and objective cognitive performance.

**Methods** Databases including PubMed, Web of Science and Cochrane Systematic Reviews were searched up to August 2017 to identify randomised controlled trials evaluating interventions for SCD. Interventions were categorised as psychological, cognitive, lifestyle or pharmacological. Outcomes of interest included psychological well-being, metacognitive ability and objective cognitive performance. To assess the risk of bias, three authors independently rated study validity using criteria based on the Critical Appraisal Skills Programme. Random-effects meta-analyses were undertaken where three or more studies investigated similar interventions and reported comparable outcomes.

**Results** Twenty studies met inclusion criteria and 16 had sufficient data for inclusion in the meta-analyses. Of these, only seven were rated as being high quality. Group psychological interventions significantly improved psychological well-being (g=0.40, 95% CI 0.03 to 0.76; p=0.03) but the improvement they conferred on metacognitive ability was not statistically significant (g=0.26, 95% CI −0.22 to 0.73; p=0.28). Overall, cognitive training interventions led to a small, statistically significant improvement in objective cognitive performance (g=0.13, 95% CI 0.01 to 0.25; p=0.03). However, the pooled effect sizes of studies using active control groups (g=0.02, 95% CI −0.19 to 0.22; p=0.85) or reporting global cognitive measures (g=0.06, 95% CI −0.19 to 0.31; p=0.66) were non-significant.

**Conclusions** There is a lack of high-quality research in this field. Group psychological interventions improve psychological well-being and may also improve metacognition. A large, high-quality study is indicated to investigate this further. There is no evidence to suggest that cognitive interventions improve global cognitive performance and the clinical utility of small improvements in specific cognitive domains is questionable. There is a lack of research considering lifestyle interventions and poor quality evidence for pharmacological interventions.

**PROSPERO registration number** CRD42017079391.

## Strengths and limitations of this study

► This is a comprehensive systematic review, including meta-analysis, of the effectiveness of psychological, cognitive, lifestyle and pharmacological interventions in subjective cognitive decline.
► Rather than solely focussing on cognitive outcome measures, this systematic review also evaluates the effectiveness of the interventions on psychological well-being and metacognition.
► The studies included in the systematic review are generally of a low quality and this makes it difficult to draw firm conclusions.

cognitive functioning, in the presence of normal performance on objective measures of cognition and no impairment of daily functioning.[1] SCD has attracted alternative names, including subjective memory impairment and subjective memory decline (SMD). The reported prevalence of SCD is variable ranging from 18% to 55% of memory clinic attendees.[2–5] However, there is no evidence-based consensus on best management for SCD and the priority of memory services is on treating patients with cognitive difficulties arising from established neurodegenerative disorders.[6]

The difficulty in treating patients with SCD stems from its aetiological heterogeneity. Broadly, two distinct groups of patients exist—those in whom SCD represents preclinical Alzheimer's disease (AD) and those in whom it does not. There is accumulating evidence for the former with Mitchell's meta-analysis[7] finding that older people with SCD were twice as likely to subsequently develop dementia. Indeed, patients presenting with SCD may already have subclinical and subtle cognitive decline suggesting an early manifestation of neurodegeneration.[8] Furthermore, there is a positive correlation between subjective cognitive complaints and amyloid deposition[9 10] and imaging studies have found brain changes indicative of AD in some patients

## INTRODUCTION

Subjective cognitive decline (SCD) describes self-reported acquired difficulties with

with SCD.[11][12] However, SCD represents a heterogenous patient group and patients without preclinical AD are less likely to demonstrate these changes.

These findings have led to growing interest in SCD, especially in the context of prodromal AD, because targeted treatments at this earlier stage could have the potential to slow or even prevent progression to AD. However, this should not detract from the reality that the majority of patients with SCD do not have preclinical AD. A recent observational study found only 14% of patients with SCD progressed to mild cognitive impairment (MCI) or AD over a 6-year follow-up[13] while a systematic review[14] highlighted that SCD symptoms frequently remit. For this non-preclinical AD group, functional psychiatric illness, personality traits, physical illness, psychosocial stress and the effects of alcohol and drug use are likely to be contributory factors.

Research into cerebrospinal fluid (CSF)[15] and neuroimaging biomarkers[9][10][16] suggest they hold promise for distinguishing between preclinical and non-preclinical AD in patients with SCD but they are not routinely used in clinical practice. Therefore, currently, for the vast majority of patients with SCD, it is not possible to make this important distinction in a timely manner and, often, only regular follow-up, over a number of years,[13][17] will reveal the underlying aetiology.

As well as identifying preclinical neurodegenerative dementias, there is an additional aspect to treating patients with SCD that must not be overlooked and this relates to the distressing symptoms that they experience. Self-perception of cognitive difficulties can lead to anger, increased stress and fear of dementia,[18] while there is growing evidence that features of anxiety and depression are common in patients with SCD.[19] Therefore, patients with SCD, regardless of aetiology, would benefit from interventions that improve psychological well-being.

Given the lack of evidence-based treatments to offer patients with SCD in the clinic, alternative options need to be sought. Metacognition, an individual's insight into their own cognitive functioning,[20] may be of relevance as a therapeutic target because, by definition, SCD involves an individual's underestimation of their cognitive ability relative to objective testing. Metternich[21] demonstrated that this patient group had significantly poorer metacognitive abilities than controls. As an overestimate of cognitive impairment may be related to difficulties in evaluating cognitive performance (ie, metacognitive difficulties), improving metacognitive ability may reduce the subjective cognitive impairment experienced by these patients.

Three systematic reviews evaluating interventions for SCD have been published.[18][22][23] Metternich and colleagues'[18] found that 'expectancy change', which included interventions such as cognitive restructuring or psychoeducation to change beliefs about cognition, was significantly more effective at improving metacognition than a non-active control intervention. A limitation of this systematic review was that 9 of the 14 studies included involved participants who were healthy volunteers and

did not necessarily have SCD. Cavenelli[22] reviewed six studies of cognitive interventions for SCD and found improvements in objective cognitive performance, often limited to the cognitive domain being trained. The most recent systematic review[23] on this topic, perhaps benefitting from being the first since the conceptualisation of the term SCD,[1] used a more rigorous inclusion criteria and found a small effect size for all non-pharmacological interventions (NPI).

## Purpose of current review

There is no consensus on the best treatment for SCD. We acknowledge the importance, both clinically and in research, of distinguishing between preclinical and non-preclinical dementia types of SCD but are mindful that this is often difficult in the majority of cases in clinical practice. Therefore, this review is a pragmatic analysis of which treatment, if any, should be offered to patients presenting with SCD in clinical settings when CSF and neuroimaging biomarker investigations are not routinely available.

In order to establish the best current treatment evidence, we aimed to conduct a systematic review of all types of interventions (psychological, cognitive, lifestyle and pharmacological) and to evaluate their effects on psychological well-being, metacognition and objective cognitive performance. Psychological well-being was chosen as an outcome measure because psychological distress is common in SCD,[18][19] metacognitive ability because it may be inversely linked to subjective cognitive complaints[21] especially in non-preclinical dementia SCD and objective cognitive performance as this has previously been identified as the main outcome in intervention studies in SCD[23] and may reflect the subtle impairment found in this patient group.

## METHODS
### Search strategy and selection criteria

We searched PubMed, Web of Science, Cochrane Systematic Reviews Database, PsycINFO and CIANHL in August 2017. Search terms used were: 'memory', 'cognitive' or 'forgetfulness' AND 'subjective', 'functional', 'benign', 'self-reported', 'complaints', 'complainer' or 'healthy' AND 'training', 'intervention', 'treatment', 'therapy' or 'medicine' (see online supplementary appendix 1 for further details on search strategy). We did not apply limits for language or date of publication. Furthermore, reference lists of included studies were manually scanned for additional relevant papers.

### Inclusion and exclusion criteria

We included studies where participants, either all or a separately analysed subgroup, had SCD defined as being self-reported cognitive complaints, without objective evidence of deficits on cognitive testing and unaccounted for by medical or psychiatric causes.[1] The intervention delivered needed to fall within the domains

of psychological, cognitive, lifestyle or pharmacological. During data extraction, three authors (RB, AJB and JDH) independently assigned the study intervention to one of these domains and when there were multiple components they judged this on the most predominant activity. Outcomes of interest included psychological well-being, metacognition and objective cognitive performance. Studies with active and non-active comparator groups were included. There were no age restrictions and only randomised controlled trials (RCTs) were included. Studies including patients with MCI or dementia were excluded as were those including patients with significant psychiatric or physical comorbidities.

## Assessment of trial quality

In order to assess the risk of bias, three authors (RB, AJB and JDH) independently rated study validity using criteria that were based on the Critical Appraisal Skills Programme (www.hello.nhs.uk/documents/CAT6-Randomised_Controlled_Trials.pdf) and have been used previously.[24 25]

Five criteria evaluating quality and bias were assessed and based on whether studies reported the following:
1. Randomisation of participants to treatment and control groups.
2. Blinding of participants and clinicians, where possible, to treatment allocation.
3. Use of 'intention to treat' analysis for incomplete outcome data.
4. Participant follow-up and data collection equal in all groups.
5. An a priori power calculation based on at least one of our outcomes of interest.

Any disagreements between raters (RB, AJB and JDH) were resolved through discussion with a fourth author (RJH). Each checklist item above scored one point; therefore, each study could score between 0 and 5 for validity. As with previous work[25] studies which independently randomised participants to treatment and control groups, used intention-to-treat analysis to manage incomplete outcome data and followed up patients in the same way (points 1, 3 and 4 above) were deemed to be high quality.

## Data extraction

The studies' demographic details, participant characteristics, main findings, limitations and results in the form of means and SD for each outcome measure were independently extracted by the three authors (RB, AJB and JDH).

## Statistical analysis

We compared intervention and control groups post-intervention using RevMan software. The programme uses Hedges adjusted g[26] to calculate a standardised mean difference (SMD) which is adjusted for small sample bias. Meta-analyses were performed where three or more studies investigated a comparable intervention and

outcome using a random effects model. Heterogeneity was quantified using the $I^2$ statistic.

Where a study reported multiple outcome measures for one outcome domain (eg, within psychological well-being), we calculated a composite measure to provide a single quantitative measure for meta-analysis.[27]

For example, if a study reported two relevant outcomes, expressed as two effect sizes ($y_1$ and $y_2$) the overall mean effect size for the composite measure will be:

$$\bar{y} = 1/2 \, (y_1 + y_2)$$

The variance of the composite effect size is calculated as follows:

$$V_{\bar{y}} = 1/4 \, (V_{y1} + V_{y2} + 2r * \sqrt{V_{y1}} * \sqrt{V_{y2}}),$$

where r is the correlation coefficient describing the extent to which $y_1$ and $y_2$ covary. In the absence of existing literature to identify a suitable correlation, we reported composite effect sizes calculated using a correlation of 0.5.

Further details of statistical methods are found in the supplementary material (see online supplementary appendix 1).

## Patient and public involvement

There was no direct patient or public involvement in this review.

# RESULTS
## Description of studies

Sixty-three potentially relevant articles were retrieved through our initial search. Of these, 20 studies met inclusion criteria; however, only 16 studies had sufficient outcome data to allow calculation of effect sizes for inclusion in meta-analyses (figure 1). The characteristics and validity of included studies are described in the online supplementary material (online supplementary data file 2: Table DS1).

Of the 20 included studies, five were classified as psychological interventions—four as group psychological treatment[28–31] and one as mindfulness training.[32] Eleven studies were classified as cognitive or memory training,[28 33–42] two as lifestyle[35 43] and four as pharmacological[44–47] interventions. Three studies[31 33 46] examined two differing interventions from the same category while two studies investigated interventions from different categories.[28 35]

Participants were community dwelling in all of the studies although the source of recruitment varied—memory clinics in two studies,[30 42] community centres in three studies[28 37 38] and the remainder through more widespread advertisement.

Of the RCTs included in the meta-analyses, 12 had a non-active control[28–30 33 36 39 41 43–47] while four had an active control.[32 35 37 40] The studies included a range of outcomes and only five RCTs[29 30 35 37 44] clearly stated their primary outcome a priori. Fifteen RCTs[28 30 32 33 35–37 39–41 43–47]

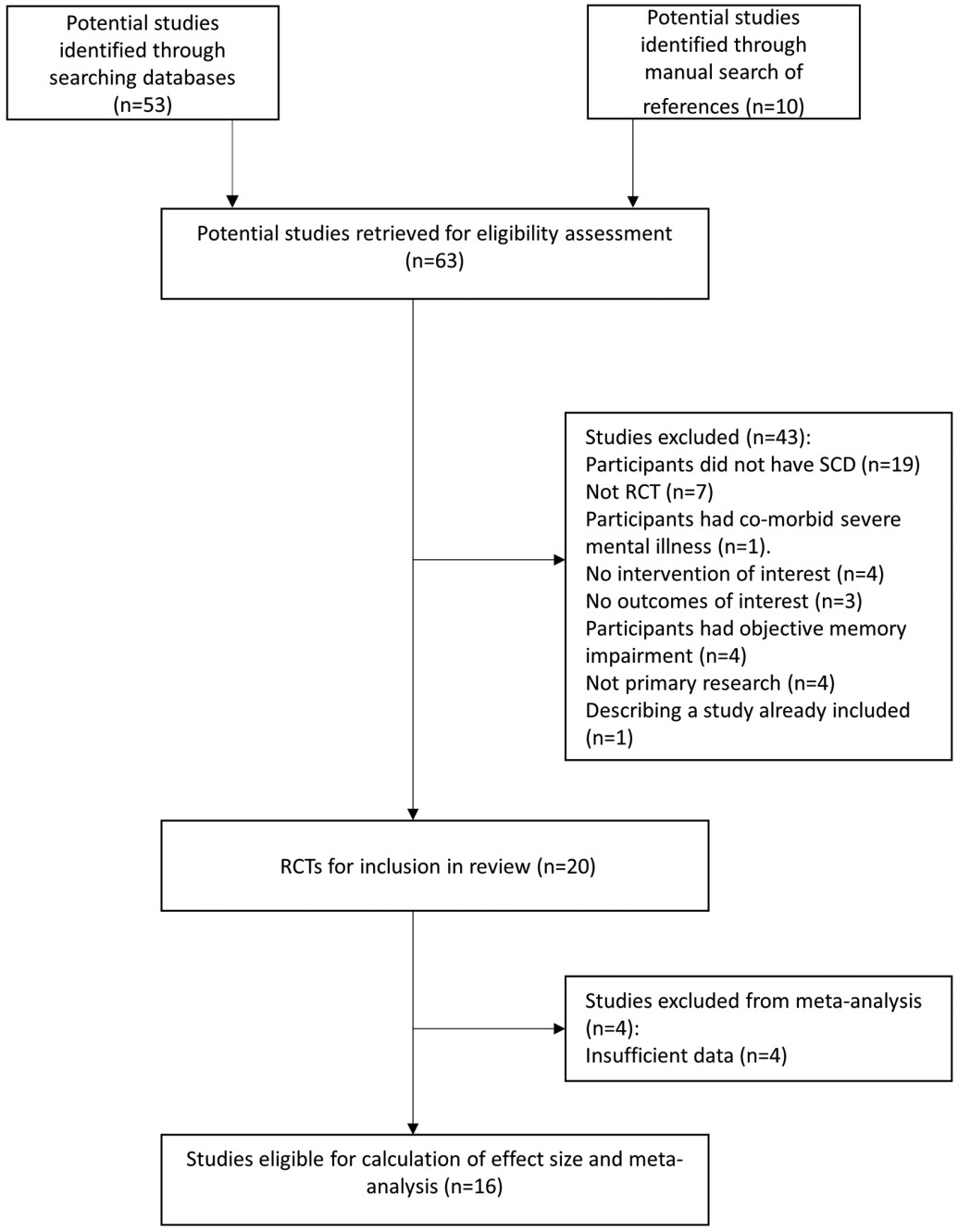

**Figure 1** Details of search strategy. RCT, randomised control trial; SCD, subjective cognitive decline.

included in the meta-analyses included objective cognitive performance among their outcomes but only three of these used an assessment of global cognitive function.[36 37 40] Ten studies[28 30–34 38 39 41 43 46 47] reported metacognitive outcomes, of which three provided a global measure of metacognition.[31 33 41] Seven studies[28 30 32 33 36 39 44] reported psychological well-being outcomes. All of the RCTs reported outcome data within 2 weeks of the intervention ending and four studies[30 39–41] collected further follow-up data up ranging from 1 to 6 months postintervention.

## Quality of studies

According to criteria described in the methods section, seven studies[29 30 35 41 43 44 47] included in the meta-analyses

were rated as being of high quality. An a priori power calculation was only carried out in two studies[30 35] while the analysis of incomplete outcome data was adequately addressed in only eight studies.[29 30 35 41 43 44 47] Thirteen[29–31 33, 35–37 39 41–47] out of 16 studies met criteria for adequate randomisation of participants and the same number of studies[29 30 32 33 35 36 39 41–47] ensured adequate blinding of participants and researchers. Publication bias was assessed using funnel plots for each outcome of interest (see online supplementary appendix 2; figure DS1).

## Group psychological interventions (four studies)

Four small studies[28–31] investigated group psychological interventions. Three[29–31] were assessed to be high quality.

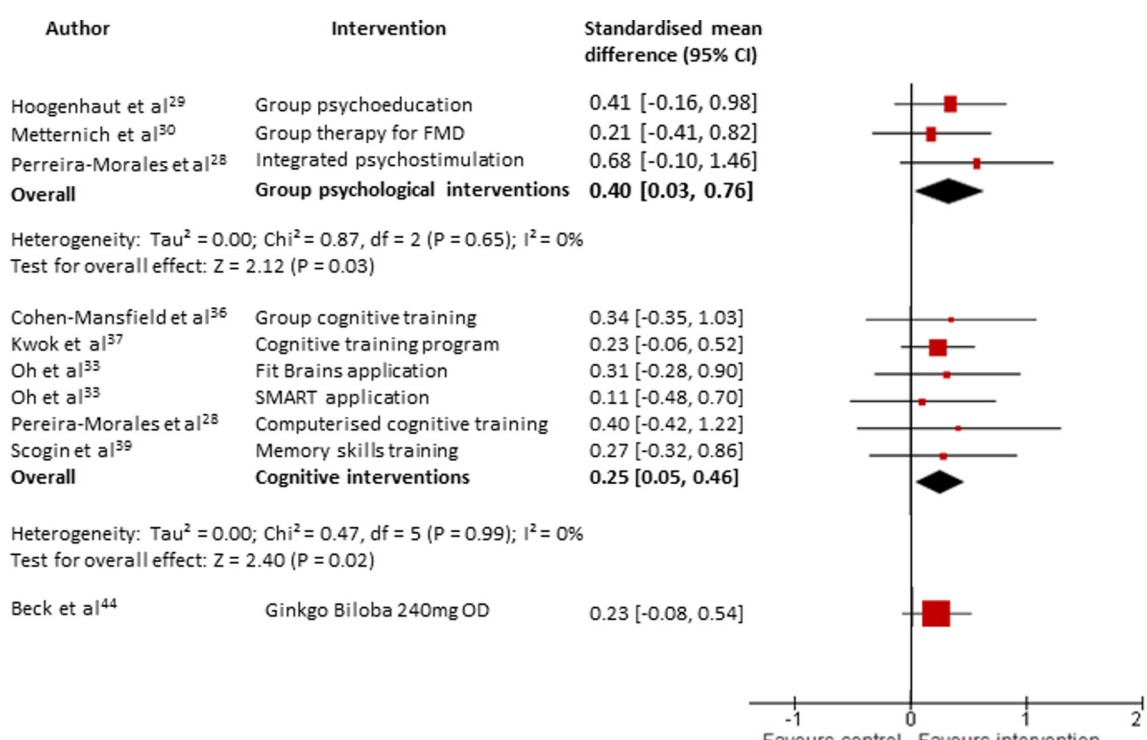

**Figure 2** Forest plot comparing interventions (all types) versus control (all types) for patients with subjective cognitive decline. Outcome: psychological well-being (at the end of intervention). FMD, Functional memory disorder; SMART, Smartphone-based brain Anti-aging and Memory Reinforcement Training.

Their main focus included psychoeducation about cognitive ageing and the link between cognitive difficulties and stress or anxiety together with goal management, strategies to improve metacognition and relaxation training. One RCT[31] which showed a reduction in anxiety symptoms in the intervention group compared with the control group could not be included in meta-analyses due to insufficient baseline data. The meta-analyses for the three remaining studies showed a significant improvement in psychological well-being postintervention (g=0.40, 95% CI 0.03 to 0.76, Z=2.12, p=0.03, total n=65, see figure 2).

When meta-analysed, group psychological interventions[28–30] conferred a positive effect on metacognition but this was not statistically significant (g=0.26, 95% CI −0.22 to 0.73, Z=1.07, p=0.28, total n=65, see figure 3). Interestingly, Metternich and coworkers[30] found that their group-based therapy did not improve memory self-efficacy (MSE) immediately post-intervention but there was a significant improvement in MSE at 3 months follow-up (p=0.04).

Three of the studies[28 29 31] measured objective cognitive performance. Hoogenhout[29] did not find any significant differences in objective measures of memory (U=229, p=0.161, n=30) or executive functioning and speed (U=274, p=0.616, n=30). Similarly, there was no effect on executive function, the only cognitive domain evaluated by Van Hooren et al.[31] In Pereira-Morales' study,[28] the Integrated Psychostimulation Programme, which contains cognitive training, conferred a significant improvement in a composite measure of objective cognitive performance (see figure 4).

### Mindfulness (one study)
Smart et al[32] conducted a pilot RCT (n=15) to assess the feasibility of mindfulness in SCD. The control group received psychoeducation and both groups demonstrated an improvement in MSE together with a reduction in cognitive complaints.

### Lifestyle interventions (two studies)
A high-quality factorial design RCT[35] showed no benefit of aerobic exercise on objective cognitive performance compared with an active control (gentle exercise that did not increase heart rate) (g=0.04, 95% CI −0.25 to 0.33, n=32, see figure 4). However, there was significant improvement in both groups.

Small's[43] small but high-quality RCT found that an intervention to promote healthy lifestyle strategies did not improve metacognition (g=−0.11, 95% CI −0.80 to 0.58, n=8, see figure 3). There was an improvement, although not reaching statistical significance, in a composite measure of verbal learning and memory and verbal fluency (g=0.42, 95% CI −0.42 to 1.26, n=8, see figure 4). The small sample size and lack of monitoring of the compliance of participants prevents any firm conclusions being drawn.

### Cognitive training interventions (11 studies)
Thirteen cognitive or memory training interventions across 11 RCTs[28 33–42] were evaluated. Three studies[34 38 42] provided insufficient data to be included in meta-analyses. Only two studies[35 41] were rated as high quality. For objective cognitive performance, significant pooled effect

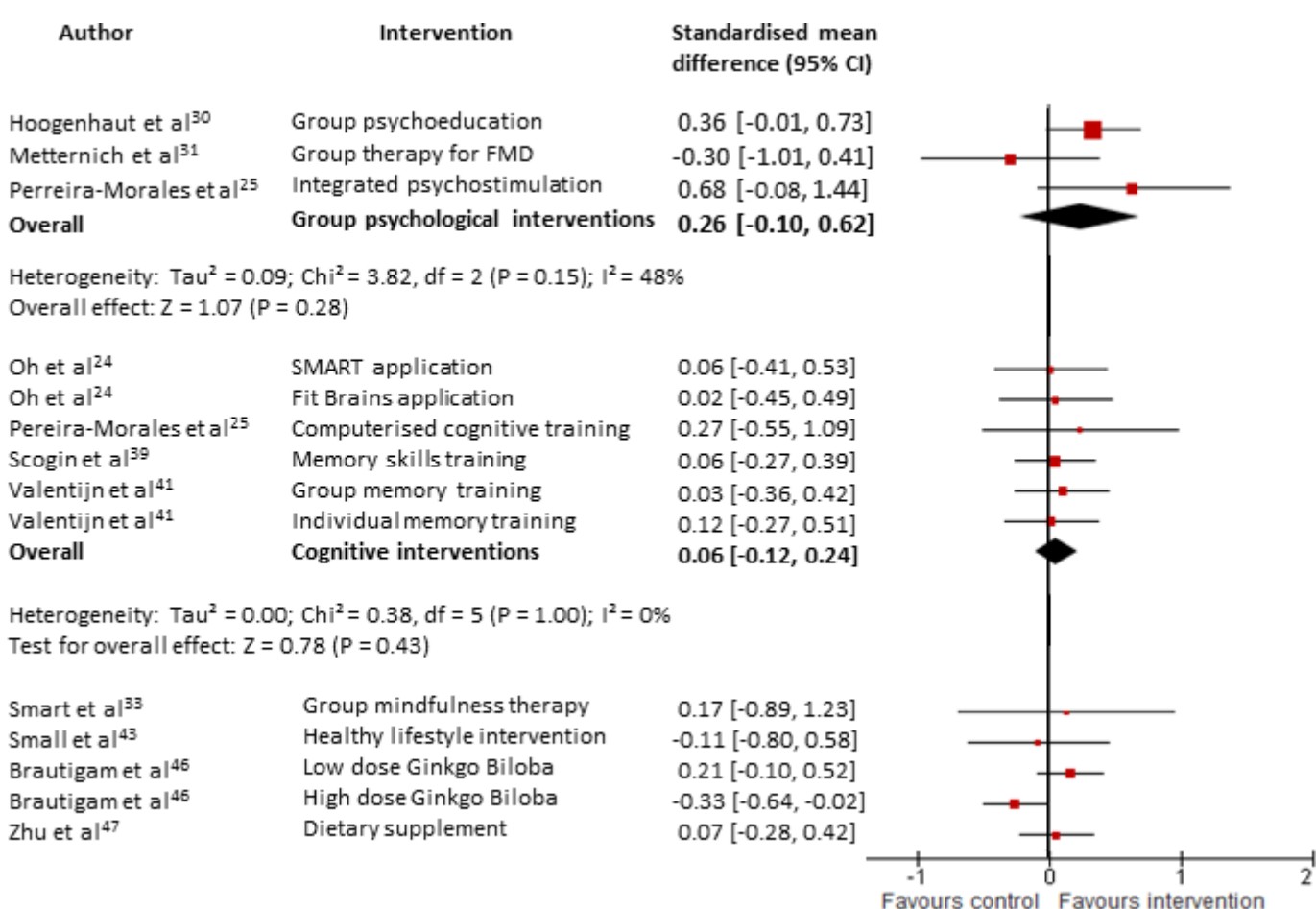

**Figure 3** Forest plot comparing interventions (all types) versus control (all types) for patients with subjective cognitive decline. Outcome: metacognitive ability (at the end of intervention). FMD, Functional memory disorder; SMART, Smartphone- based brain Anti-aging and Memory Reinforcement Training.

sizes were found (g=0.13, 95% CI 0.01 to 0.25, Z=2.13, p=0.03, total n=311, see figure 4) with no heterogeneity (I²=0%). Three studies[35 37 40] had active control groups in which participants received forms of cognitive stimulation and when analysed separately, their pooled effect size was equivocal (g=0.02, 95% CI –0.19 to 0.22, Z=0.19, p=0.85, total n=132, see figure 5). Conversely, for the remaining studies which used non-active controls, the pooled effect size was significant (g=0.18, 95% CI 0.04 to 0.33, Z=2.48, p=0.01, total n=179, see figure 5).

Three studies[36 37 40] used global cognitive performance outcome measures and when meta-analysed, there was no discernible improvement postintervention (g=0.06, 95% CI=–0.19 to 0.31, Z=0.45, p=0.66, total n=114, see online supplementary appendix 2: figure DS2).

Five studies[28 33 36 37 39] reported psychological well-being. Each of the studies demonstrated an improvement in psychological well-being but individually none were statistically significant. However, when meta-analysed, there was a small beneficial effect on psychological well-being (g=0.25, 95% CI 0.05 to 0.46, Z=2.40, p=0.02, total n=169, see figure 2) with no heterogeneity (I²=0). Three studies[33 39 42] reported metacognitive outcomes and pooled together there was no significant effect (g=0.06,

95% CI –0.12 to 0.24, Z=0.66, p=0.51, total n=165, see figure 3).

### Pharmacological interventions (four studies)

Two studies investigated the effectiveness of Ginkgo biloba. Brautigam's low-quality but relatively large trial[46] included two different dosing regimens, namely high-dose and low -dose Ginkgo extract for 24 weeks. They led to conflicting results with high-dose Ginkgo causing a statistically significant worsening in metacognition while low-dose Ginkgo produced the opposite effect although statistically insignificant.

In Beck and colleagues' trial,[44] a 240 mg preparation of Ginkgo biloba was administered to the intervention group daily for 8 weeks. Both psychological well-being and objective cognitive performance improved but neither was statistically significant.

One small, low-quality trial by Boespflug et al[45] found that 2.4 g fish oil daily had no effect on working memory.

Zhu and colleagues[47] evaluated the effectiveness of BrainPower Advanced, a dietary supplement, taken two times per day for 12 weeks. The trial was of a poor quality with marked baseline differences in severity of cognitive complaints between groups. Analysing the primary data,

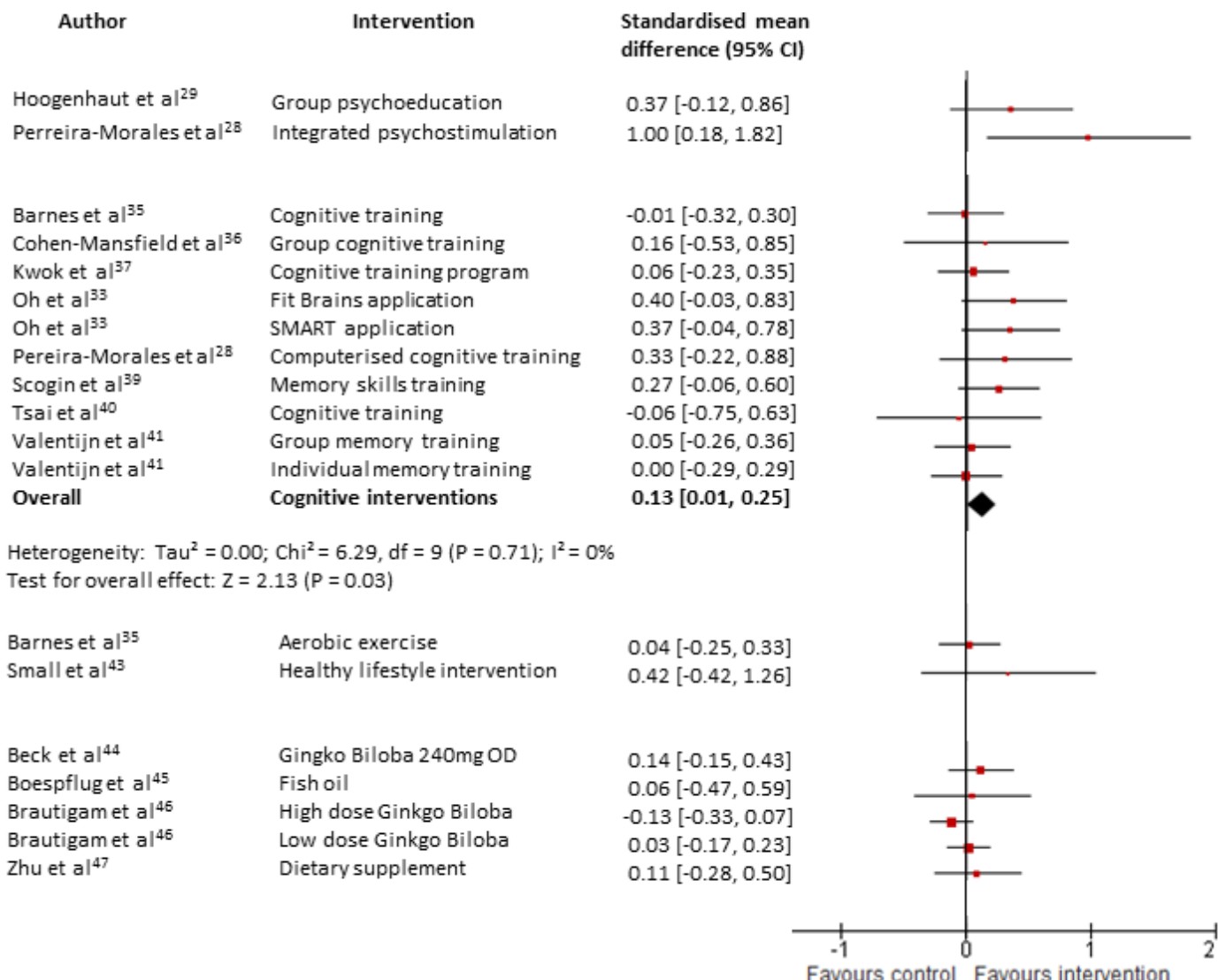

**Figure 4** Forest plot comparing interventions (all types) versus control (all types) for patients with subjective cognitive decline. Outcome: objective cognitive performance (at the end of intervention). SMART; Smartphone-based brain Anti-aging and Memory Reinforcement Training.

there was no significant difference in metacognitive ability (g=0.07, 95% CI −0.35 to 0.31, n=47, see figure 3) or objective cognitive performance (g=0.11, 95% CI −0.28 to 0.50, n=47, see figure 4) between the groups.

## DISCUSSION
### Main findings
The most obvious finding from our review is the lack of good quality evidence in this field. However, given that this is an emerging field, we attempted to be as inclusive as possible and included low-quality studies in our meta-analyses provided they did not have an inadequate or unclear rating in all five areas of bias that we assessed (see methods).

There was a statistically significant improvement in psychological well-being for interventions which incorporated psychoeducation and were delivered in group settings, although the effect size was only moderate

(g=0.40).[48] The studies, two of which were of high quality, are limited by small sample sizes and one of the interventions[28] included cognitive training in addition to a psychological approach. Overall, however, this could be a promising intervention that can be feasibly implemented into clinical practice.

We found that cognitive interventions conferred a small but statistically significant effect (g=0.13) on cognitive outcomes which is in keeping with Smart's earlier review.[23] However, the clinical relevance of such a small improvement is easy to question. The apparent clinical utility of cognitive interventions in SCD is further undermined by our finding that meta-analysis of the three studies which used global cognitive performance outcome measures found no overall benefit. Our use of composite measures provided us with a single effect size for cognitive performance in each study but we are mindful that this merely reflects performance across the cognitive

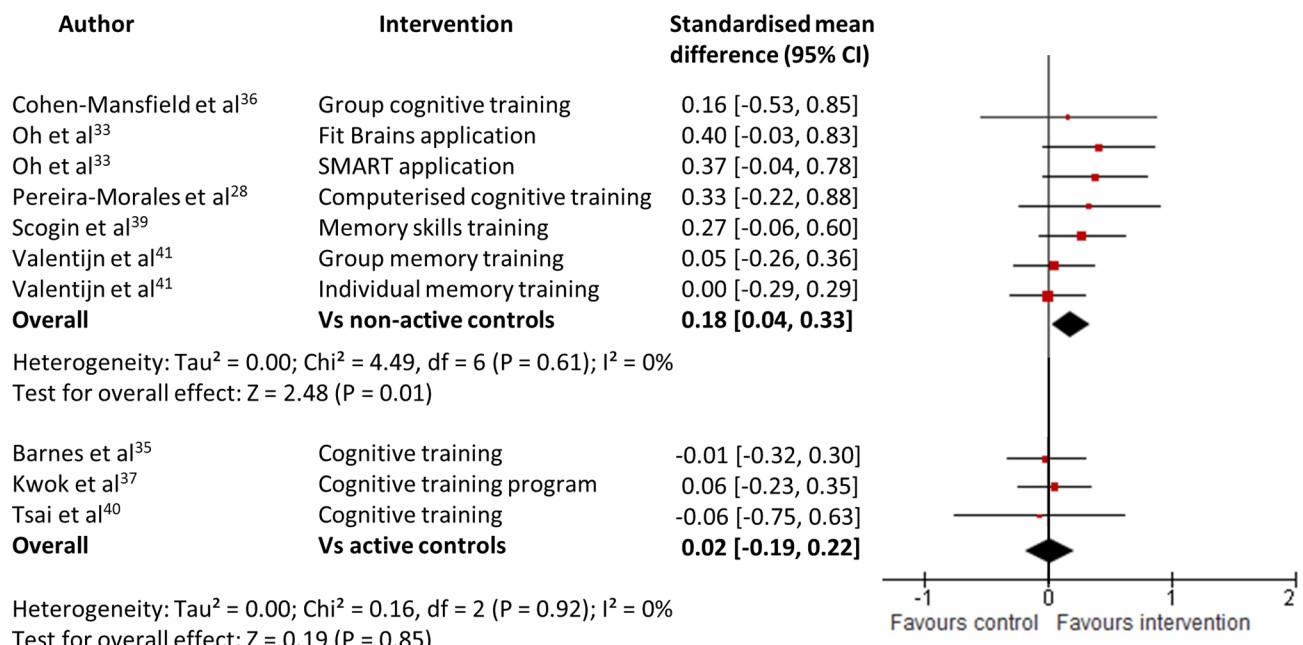

| Author | Intervention | Standardised mean difference (95% CI) |
|---|---|---|
| Cohen-Mansfield et al[36] | Group cognitive training | 0.16 [-0.53, 0.85] |
| Oh et al[33] | Fit Brains application | 0.40 [-0.03, 0.83] |
| Oh et al[33] | SMART application | 0.37 [-0.04, 0.78] |
| Pereira-Morales et al[28] | Computerised cognitive training | 0.33 [-0.22, 0.88] |
| Scogin et al[39] | Memory skills training | 0.27 [-0.06, 0.60] |
| Valentijn et al[41] | Group memory training | 0.05 [-0.26, 0.36] |
| Valentijn et al[41] | Individual memory training | 0.00 [-0.29, 0.29] |
| **Overall** | **Vs non-active controls** | **0.18 [0.04, 0.33]** |

Heterogeneity: Tau² = 0.00; Chi² = 4.49, df = 6 (P = 0.61); I² = 0%
Test for overall effect: Z = 2.48 (P = 0.01)

| | | |
|---|---|---|
| Barnes et al[35] | Cognitive training | -0.01 [-0.32, 0.30] |
| Kwok et al[37] | Cognitive training program | 0.06 [-0.23, 0.35] |
| Tsai et al[40] | Cognitive training | -0.06 [-0.75, 0.63] |
| **Overall** | **Vs active controls** | **0.02 [-0.19, 0.22]** |

Heterogeneity: Tau² = 0.00; Chi² = 0.16, df = 2 (P = 0.92); I² = 0%
Test for overall effect: Z = 0.19 (P = 0.85)

**Figure 5** Forest plot comparing cognitive interventions versus non-active controls and active controls in separate analyses for patients with subjective cognitive decline. Outcome: objective cognitive performance (at the end of intervention).

domains assessed and is not a surrogate for global cognition. Smart[23] suggested that the small improvements in cognition are important because they demonstrate that cognitive enhancement and compensation is possible in this patient group. While this may be of relevance when considering the small but significant minority of persons in whom SCD represents preclinical AD, this finding has less bearing on the remaining majority who, by definition, have normal objective cognitive performance.

We found that cognitive training conferred a small but significant benefit on psychological well-being. The effects of cognitive training on psychological well-being have been poorly researched[49] and there is sometimes an unproven assumption that cognitive training enhances a patient's overall well-being by preserving cognitive function.[50] This would not explain our findings given patients were evaluated within a short time frame from baseline. The larger effect on psychological well-being than on objective cognitive performance measures may reflect a significant psychogenic aetiology in many cases of SCD. However, it is important to note that the evidence is of a poor quality and this may represent a chance finding given that psychological well-being was not a primary outcome for any of the cognitive training intervention studies.

Metacognitive deficits are relevant in subjective cognitive disorders.[21] Psychoeducation delivered in group settings did not lead to a significant improvement in metacognitive ability immediately after the intervention but this finding was influenced by Metternich and colleagues'[30] study which found less improvement in the intervention group compared with the control (g=−0.30, 95% CI −1.01 to 0.41). However, their intervention led to a significant improvement in metacognitive ability

when assessed 3 months post-treatment leading them to postulate that a period of time is needed for cognitive restructuring and the implementation of strategies into everyday life. Our findings suggest that further work is required to explore the role of group psychoeducation in improving metacognition and longer lengths of follow-up are indicated. A limitation to the literature investigating metacognitive outcomes is the broad range of subjective measures in use.

There is only limited evidence available with regards to lifestyle interventions for SCD. Exercise was evaluated in one high-quality RCT. The intervention group performed aerobic exercise while the control group performed gentle exercise designed not to increase heart rate. Both groups showed significant improvements in objective cognitive performance but there was no difference between the groups. This may suggest that exercise, regardless of intensity, could be beneficial in SCD and further RCTs with a non-active control group are indicated.

Five studies have investigated alternative medicines and medicinal supplements in SCD. Overall, the evidence for their efficacy is not encouraging. One study[44] found that 240 mg daily of Ginkgo biloba for 8 weeks led to non-significant improvements in psychological well-being and cognitive performance. Brautigam and colleagues[46] used two alternative doses of Ginkgo biloba and their results are conflicting. Higher doses led to significantly worse metacognition compared with control and lower doses led to statistically insignificant improvements. Fish oil[45] did not improve objective cognitive performance while a dietary supplement[47] did not improve either metacognition or objective cognitive performance.

In summary, psychoeducation delivered in group settings appears to be effective at improving psychological

well-being, which is a relevant and important outcome, in SCD.[18 19] However, individually none of the three studies demonstrated statistically significant improvements which was largely due to their small sample sizes. A large RCT is urgently needed to extrapolate our findings for this clinically practical intervention which is highly promising. Metacognition, an individual's insight into their own cognitive functioning, is a potential area of research in SCD because deficits in this area may contribute to the condition, especially the non-preclinical dementia type. Specific interventions to improve metacognitive performance as well as to objectively quantify it are emerging[20] and these may be of relevance in SCD. Cognitive interventions have a modest effect in improving objective cognitive performance and such small improvements are unlikely to be of clinical significance for the vast majority of patients with SCD who are not at risk for developing dementia. There is limited research considering lifestyle interventions in SCD and only unconvincing and poor quality evidence for pharmacological interventions in SCD.

## Limitations

There is a high variability in measurement instruments used to define SCD in studies[51] which limits reviews such as our own. Furthermore, SCD is a construct which includes a heterogenous group of patients that could be broadly divided into two distinct patient groups—those with and without preclinical dementia. All of the studies in our review included both groups of SCD patients together and this limits our ability to make inferences about targeted interventions for each group. Future work should investigate interventions for each group of patients separately if possible.

The poor quality of studies in this review due to a lack of clearly defined primary outcomes and small sample sizes has been highlighted previously and limits the confidence with which we can draw conclusions. Funnel plots assessing publication bias (see online supplementary appendix 2; figure DS1) appeared asymmetrical. Small sample sizes and heterogeneous interventions may be contributory factors but we are mindful that when the number of studies in the meta-analysis is small as is the case in this review, such asymmetry may arise by chance rather than reflect true publication bias.[52]

The interventions were heterogenous. As well as differing formats and structures of interventions which we attempted to group into broad categories, there were differences in lengths of intervention, monitoring of compliance and the motivation of participants which are more difficult to quantify. Often interventions had multiple components and we had to determine which domain was the most prominent. This may mean the most effective component of the intervention was not appropriately captured.

Similarly, the studies included in the meta-analyses contained varied control groups. In at least two[32 40] of the four studies using an active control group, the control group received treatment which may have been effective in its own right thereby masking the possible effectiveness of the intervention. The other studies used non-active control groups but there is likely to have been much variability in what treatment as usual constituted.

The goal of research into interventions must be for them to be translational into clinical practice. Unfortunately, only one study[30] included in the meta-analyses recruited patients from a healthcare setting while the remainder relied on convenience sampling of volunteers with subjective cognitive complaints from the community. This may limit how translatable our findings are to clinical populations.

## Implications

This review demonstrates the need for higher quality studies in SCD research with a distinction made between SCD in preclinical AD and otherwise. The development of a conceptual framework for research in SCD[1] should aid this. This review would support high-quality RCTs of a group psychological intervention targeting psychological well-being and metacognition in patients with SCD. Follow-up data at least 3 months postintervention should be collected. Metacognitive ability may be an important therapeutic target in SCD and interventions to directly improve this along with objective measurement tools are required.

**Contributors** All authors made a substantial contribution to this work. RB, AJB, JDH and RJH all contributed to the conception and design of the review. RB read and screened abstracts and titles of potentially relevant studies. RB, AJB and JDH read the retained papers and were responsible for extracting data and rating their quality independently. RB drafted the paper with all the authors critically reviewing it and suggesting amendments prior to submission. All the authors had access to all the data in the study and can take responsibility for the integrity of the reported findings.

**Funding** This research received no specific grant from any funding body. RB is supported by a NIHR Academic Clinical Fellowship. JH and RH are supported by the NIHR UCLH BRC.

**Competing interests** None declared.

**Patient consent** Not required.

**Provenance and peer review** Not commissioned; externally peer reviewed.

**Data sharing statement** Details of excluded papers are available from the first author on request.

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
