## [Reviewer comments · BMJ Open]

ARTICLE DETAILS

TITLE (PROVISIONAL)	Interventions for Subjective Cognitive Decline: Systematic Review and Meta-analysis
AUTHORS	Bhome, Rohan; Berry, Alex; Huntley, Jonathan; Howard, R

VERSION 1 – REVIEW

REVIEWER	Rachel Buckley Massachusetts General Hospital, United States
REVIEW RETURNED	12-Feb-2018

GENERAL COMMENTS	The authors present a meta-analysis on interventions for SCD to improve well-being, metacognition and objective cognitive performance. While this is a statistically appropriate review of the research question posed, the authors do not present a comprehensive argument for the utility of intervention of SCD. Jessen and colleagues from the SCD Initiative largely refer to SCD as an operationalization of a clinical marker of preclinical Alzheimer's disease. While there is research out there that tackles questions of intervention for SCD, the authors do not make it clear what the intention is for treatment. That is, if this is a clinical marker for pathological burden, is the intention to alter the course of the AD pathophysiologic trajectory? Clearly, this is not the case, as the authors pose the intent to examine the issue of changes in well-being, metacognition, etc, post-treatment of SCD. If this is the case, however, what might be the implications of intervention of the symptoms of SCD for AD pathology? One natural extension of this thought is: if one intervenes successfully for the SCD symptomatology, does this now mask the ever-encroaching pathological burden, thus making it more difficult to detect and track over time? As such, my question as a whole for the SCD intervention literature (which the authors have meta-analyzed, and thus can speak to this with some authority) is: what is the overall goal of SCD intervention, and how is this important for the biological aspect of preclinical Alzheimer's disease? This is a very salient issue in my mind: clearly, better well-being and overall quality of life for individuals in the preclinical phase of AD is extremely important, but it is not clear to me that this is the focus of the meta-analysis (i.e. those with preclinical AD vs just those with SCD). If the focus is on just those with SCD (which is arguably a very heterogeneous group of those in the preclinical AD phase, and those who are manifesting un-related or parallel-related pathologies/conditions), then treating all-comers becomes a very confusing goal. Some comment from the authors on this important topic, both alluded to in the introduction, and as a discussion point in the manuscript, is highly pertinent. 1. In the abstract, how do the authors confer an effect of improved well-being on metcognitive ability when there is no significant effect
---

	in their analyses? 2. In relation to FMD: The authors seem to be referring to MCI, not SCD in the operationalised sense, so it is important to make it clear what you are referring to. For instance, what is a functional memory disorder, does this mean memory impairment? This is not the operationalisation of SCD, in fact, SCD is defined by normal objective memory performance in the face of subjective concern. 3. At the point when the authors state: “Effective interventions for patients with SCD are necessary.” In reference to the comments above - why is this the case? What is the intended treatment effect and outcome here? And the underlying mechanism? One could argue that interventions of those with BIOLOGICAL EVIDENCE OF AD is more important. And then measuring the changes in objective cognitive performance is a primary outcome with changes in SCD is secondary to that. A further discussion of the author’s rationale is important here. 4. When the authors state: “For those with pre-clinical AD the significant social, personal and economic impact of dementia¹⁶ make treatments that could slow or even prevent progression to AD particularly important.” This is a critical statement that is not clearly linked with the previous points that the authors make. Is the intention of treating SCD to slow or prevent the biological mechanisms of AD? If this is the case, this is an extremely long bow to draw. It is unlikely that treating clinical symptoms of subjective cognitive decline will alter the AD pathophysiological pathway. As such, the authors should consider an alternative statement here, or a further elucidation of their argument. 5. Line 35 from page 3 in reference to “anger and stress”: While, some studies may report this, this has not always been reported by qualitative studies of SCD by Clare et al., or Buckley et al., who show that phenomenological experiences of subjective concerns are more related to anxiety, concern, and changing insight. Not anger or stress, per se. These are very strong qualitative terms that are not relayed in the qualitative literature to my knowledge. 6. The authors need to explain the rationale of the paragraph starting at line 41 on page 3, which relates back to the earlier points that have been made: if metacognitive abilities are affected in SCD without evidence of preclinical AD, what is the intention of the intervention for SCD? Is it to reduce SCD in those who shouldn’t be showing it, or is there perhaps some other cause that is driving the SCD? Please make this clearer. 7. Further, is metacognition affected in AD, particularly in the preclinical phase when SCD is of particular salience? If not, what is the point of this intervention with reference to metacognition? This point is somewhat lost! 8. In relation to the argument on line 25 (page 8), this sort of control group in a clinical trial is not a limitation, as it allows for a comparison over and above a ‘normal’ intervention. It is very hard to ensure a true ‘negative’ control group is ‘negative’. 9. Please include a funnel plot so that the reader can see whether there is publication bias in the effect sizes reported in these studies. 10. Line 50, page 10: There is minimal evidence to support metacognitive issues in SCD - indeed, the field does not focus on this area very much to my knowledge. The assertive tone of this argument should be muted somewhat. 11. Line 54 of page 11: there are many biomarkers that can be used to investigate preclinical AD - particularly with regards to CSF and PET neuroimaging. And APOE genotyping is often done in studies -
--	--

	it is unclear what the authors are referring to here?
REVIEWER	Sietske Sikkes Alzheimer Center, VU University Medical Center, the Netherlands
REVIEW RETURNED	08-Mar-2018

GENERAL COMMENTS	Review Interventions for Subjective Cognitive Decline: Systematic Review and Meta-analysis This is a timely and interesting systematic review focusing on interventions for SCD, which is a target population of growing interest, because of their potential higher risk of developing dementia. Despite the relevance for this target populations, I do have some concerns regarding the methodology and the conclusions drawn by the authors.  1. First of all, the focus of the paper is unclear: is this aimed at SCD in the context of AD or is it focused on SCD as a 'functional memory disorder'? Both terms are used in the manuscript, and have very different implications for which studies should be included in the review and the literature that should be discussed. Related to that, it is unclear why the authors chose to combine all potential interventions into one review, for complementary medicine it is noted that 'this has not been evaluated before', which does not seem a legitimate reason to fit it in the current study. From the introduction, it is not clear what the current study is adding to previous systematic reviews for this population. 2. The literature included in this paper is insufficient: SCD in the context of AD is missing some highly relevant citations on the definition and operationalization of SCD that resulted from the SCD working group (e.g. Molinuevo et al. 2017); as well as literature regarding the potential to prevent AD in SCD. Also regarding the concept of 'metacognition' which is mentioned as a potential cause for SCD the literature is not adequately discussed; this could indeed be a cause, but the authors do not discuss relevant literature indicating that SCD is preceded by actual cognitive decline (e.g. Koppa et al. 2015). 3. An important issue lacking in the discussion is the definition of SCD, which may differ by differences in populations (e.g. memory clinic versus population-based), but also by means of the high variability in measurement instruments that are used to define SCD (Rabin et al. 2015). 4. The authors chose a very broad approach, looking at all potential interventions in all possible SCD populations (population-based vs memory-clinic based). This is comparing apples to oranges, and severely limits the usefulness of this paper. The authors should incorporate these limitations into the discussion. 5. The authors should be more cautious in their conclusions: they note on the one hand that there is a lack of high-quality evidence (RCT's) for interventions on SCD - but on the other hand they conclude that several interventions have no effect. If there's insufficient evidence, this cannot be concluded based on this study. 6. The authors state that they included all studies with SCD as a target population "although most studies used other nomenclature" - what nomenclature was used, and how did the authors define SCD? 7. Important details regarding the methods are not available: The majority of the methods section (e.g. quality assessment, creating composite scores) is in the appendix, not in the main paper - making
---

	it difficult to interpret the quality of this systematic review. In addition, how did the authors decide an intervention was 'comparable'? How did the authors decide on the target population? This information is missing from the paper. 8. The authors a priori put much weight on a 'global cognitive measure' - why is that? Would the aim not be to reduce SCD? 9. The references to functional memory disorder do not seem to relate to the concept of SCD. 10. The authors highlight important differences between memory-clinic SCD and population-based SCD in the introduction; so grouping interventions over those populations does not seem useful. This should be part of the discussion.
--	---

REVIEWER	Andrea Tales Swansea University, Wales, UK
REVIEW RETURNED	13-Mar-2018

GENERAL COMMENTS	Please can the authors consider the following points. On page 2, in the conclusions section of the abstract the authors state that 'group psychological interventions improve psychological well-being and are likely to improve metacognition' but do the results warrant such a statement. My concern is that the analysis outcome is not robust enough, in terms of the number and quality of the studies, to claim that group psychological interventions are likely to improve metacognition ...how is 'likely' defined? Page 2: Under strengths and limitations of this study: What do the authors mean by the statement that interventions on psychological well-being and metacognition as it a more naturalistic outcome? What does naturalistic mean? Page 3. In terms of SCD prevalence, some of these references [4-7] are quite old. Is there newer data on this; especially as interest in SCD has increased since the publication of these papers? Can the use of the word 'sufferers' (page 3) be avoided. I'm a little confused by your description of SCD in the introduction on page 3. Do all people with SCD have increased amyloid deposition in their brains? What about for those for whom SCD does not represent prodromal AD or other dementia? The section on metacognition in the introduction is rather confusing; especially the second sentence and not sure of the context with respect to the present systematic review...more detail and context would be appropriate here. In the section on purpose of current review...again the term 'naturalistic' is used but without a detailed description of what this means and its context. I would also question the statement that deficits in metacognition may underpin SCD...is SCD as specific as this...More information is needed as to why these outcomes were chosen, i.e., biological/behavioural basis, and are these likely to occur in isolation? What about the fact that some SCD may be caused by medical issues or dementia and how does this fact relate to the outcome choices made for this review? Could this be linked to the decision to also include pharmacological interventions. In the methods section the authors state that participants had SCD. How was this judged bearing in mind the issue of nosology in SCD?
--

	How many participants in total were included in the meta analysis? On page 8 under the 'mindfulness' section please can the authors describe fully why they state that key limitation to Smart et al's study was that the control group received psycho-education....surely this is common practice when comparing different groups and the specificity of an intervention to disease or disorder? Discussion The word 'that' is included twice on line 2 under main findings. Paragraph 2; the authors state that, even though the effect size was only moderate, psycho-education is a promising cost effective intervention...The outcome was based only on relatively few studies. Does this evidence warrant such a statement? I feel that there is a general bias throughout this paper with respect to the description of the aetiology of SCD in relation to a behavioural/ psychogenic aetiology; the issue of the effect on results of people for whom SCD represents prodromal dementia is not much discussed. How, in the papers included in the meta analysis, SCD (or what is was called) was diagnosed is not described for all studies. On page 12 under the heading 'Implications' the authors state that this review demonstrates the need for higher quality studies in SCD research with a distinction made between SCD in pre-clinical AD and otherwise' but in what way does the review demonstrate this and how can one determine for whom SCD represents pre-clinical AD?
--	---

REVIEWER	Nick Garrett AUT, Auckland, New Zealand
REVIEW RETURNED	10-Apr-2018

GENERAL COMMENTS	There are two minor issues:  - page 7 lines 41-47 the criteria are relevant but as we are dealing with articles the issues are that a priori power calculations and randomisation are not reported rather than not carried out, suggest minor rewording to demonstrate this. - the written results section is ordered by intervention type, whereas the figures referred to are ordered by outcome. It makes it very confusing to follow. Suggest either more text in results section to clarify or to make sure the results and figures correspond.
---

VERSION 1 – AUTHOR RESPONSE

Reviewer: 1

Reviewer Name: Rachel Buckley

Institution and Country: Massachusetts General Hospital, United States Competing Interests: None

The authors present a meta-analysis on interventions for SCD to improve well-being, metacognition and objective cognitive performance. While this is a statistically appropriate review of the research question posed, the authors do not present a comprehensive argument for the utility of intervention of SCD. Jessen and colleagues from the SCD Initiative largely refer to SCD as an operationalization of a

clinical marker of preclinical Alzheimer's disease. While there is research out there that tackles questions of intervention for SCD, the authors do not make it clear what the intention is for treatment. That is, if this is a clinical marker for pathological burden, is the intention to alter the course of the AD pathophysiologic trajectory? Clearly, this is not the case, as the authors pose the intent to examine the issue of changes in well-being, metacognition, etc, post-treatment of SCD. If this is the case, however, what might be the implications of intervention of the symptoms of SCD for AD pathology? One natural extension of this thought is: if one intervenes successfully for the SCD symptomatology, does this now mask the ever-encroaching pathological burden, thus making it more difficult to detect and track over time? As such, my question as a whole for the SCD intervention literature (which the authors have meta-analyzed, and thus can speak to this with some authority) is: what is the overall goal of SCD intervention, and how is this important for the biological aspect of preclinical Alzheimer's disease? This is a very salient issue in my mind: clearly, better well-being and overall quality of life for individuals in the preclinical phase of AD is extremely important, but it is not clear to me that this is the focus of the meta-analysis (i.e. those with preclinical AD vs just those with SCD). If the focus is on just those with SCD (which is arguably a very heterogeneous group of those in the preclinical AD phase, and those who are manifesting un-related or parallel-related pathologies/conditions), then treating all-comers becomes a very confusing goal. Some comment from the authors on this important topic, both alluded to in the introduction, and as a discussion point in the manuscript, is highly pertinent.

1. In the abstract, how do the authors confer an effect of improved well-being on metcognitive ability when there is no significant effect in their analyses?

We have said group psychological interventions may improve metacognitive ability. There is no significant effect overall because in the Metternich et al, 2010 study- the effect immediately post intervention was negative. We meta-analysed studies based on effect immediately post intervention. However, when patients were followed up 3 months later the intervention was shown to be effective. We have discussed reasons why the effect may not be immediate in the paper in the discussion. Therefore, we feel group psychological interventions may be effective in improving metacognitive ability.

2. In relation to FMD: The authors seem to be referring to MCI, not SCD in the operationalised sense, so it is important to make it clear what you are referring to. For instance, what is a functional memory disorder, does this mean memory impairment? This is not the operationalisation of SCD, in fact, SCD is defined by normal objective memory performance in the face of subjective concern.

Thank you, we agree that this is misleading and FMD may represent a different population.

3. At the point when the authors state: "Effective interventions for patients with SCD are necessary." In reference to the comments above - why is this the case? What is the intended treatment effect and outcome here? And the underlying mechanism? One could argue that interventions of those with BIOLOGICAL EVIDENCE OF AD is more important. And then measuring the changes in objective cognitive performance is a primary outcome with changes in SCD is secondary to that. A further discussion of the author's rationale is important here.

Thank you. We have removed this sentence as we agree that it is unhelpful. We hope that paragraphs 5 and 6 (lines 72-84) explain why treatment for SCD is important and what the potentially important outcomes are.

We do not disagree that interventions for patients with biological evidence of AD are crucial but we have tried to make the point that we are trying to pragmatically evaluate what options are available for clinicians when faced with a patient with SCD when they do not have immediate recourse to CSF, imaging biomarker investigations and so do not have a means of discerning between preclinical AD

SCD and non-preclinical AD SCD. Please see paragraph 4 of the introduction (lines 67-91).

4. When the authors state: “For those with pre-clinical AD the significant social, personal and economic impact of dementia¹⁶ make treatments that could slow or even prevent progression to AD particularly important.” This is a critical statement that is not clearly linked with the previous points that the authors make. Is the intention of treating SCD to slow or prevent the biological mechanisms of AD? If this is the case, this is an extremely long bow to draw. It is unlikely that treating clinical symptoms of subjective cognitive decline will alter the AD pathophysiological pathway. As such, the authors should consider an alternative statement here, or a further elucidation of their argument.

Thank you. We agree with the reviewer’s comments and have removed this statement as it does not link with the previous points.

5. Line 35 from page 3 in reference to “anger and stress”: While, some studies may report this, this has not always been reported by qualitative studies of SCD by Clare et al., or Buckley et al., who show that phenomenological experiences of subjective concerns are more related to anxiety, concern, and changing insight. Not anger or stress, per se. These are very strong qualitative terms that are not relayed in the qualitative literature to my knowledge.

We have now added to this part by mentioning features of anxiety and depression are common. ‘anger and stress’ were reported in an earlier referenced study.

6. The authors need to explain the rationale of the paragraph starting at line 41 on page 3, which relates back to the earlier points that have been made: if metacognitive abilities are affected in SCD without evidence of preclinical AD, what is the intention of the intervention for SCD? Is it to reduce SCD in those who shouldn’t be showing it, or is there perhaps some other cause that is driving the SCD? Please make this clearer.

Thank you and apologies that this was unclear. We have tried to make it clearer that improving metacognition could be important for patients with non-preclinical AD SCD and these patients are commonly seen in memory clinics. The aim would be to reduce SCD in those who shouldn’t be showing it and so this has been made clearer in the 6th paragraph of the introduction. As an over-estimate of cognitive impairment may be related to difficulties in evaluating cognitive performance (i.e. metacognitive difficulties), improving metacognitive ability may reduce the subjective cognitive impairment experienced by these patients.

7. Further, is metacognition affected in AD, particularly in the preclinical phase when SCD is of particular salience? If not, what is the point of this intervention with reference to metacognition? This point is somewhat lost!

Research into metacognition in preclinical AD is limited. We would argue that SCD even when not preclinical AD is also important and in this group, who form the majority of patients with SCD accessing healthcare services, metacognitive deficits may play a role in their SCD symptoms. Therefore, improving metacognition could be important for patients with non-preclinical AD SCD.

8. In relation to the argument on line 25 (page 8), this sort of control group in a clinical trial is not a limitation, as it allows for a comparison over and above a ‘normal’ intervention. It is very hard to ensure a true ‘negative’ control group is ‘negative’.

We agree and have removed this comment.

9. Please include a funnel plot so that the reader can see whether there is publication bias in the effect sizes reported in these studies.

We have now included a funnel plot in the supplementary methods and interpreted it in the discussion under the 2nd paragraph in limitations.

We have included a funnel plot as requested by reviewer 1 (point 9). However, guidance on publication bias would suggest it is inappropriate to investigate the asymmetry when there are fewer than 10 studies per meta-analyses due to the possibility of chance findings.

10. Line 50, page 10: There is minimal evidence to support metacognitive issues in SCD - indeed, the field does not focus on this area very much to my knowledge. The assertive tone of this argument should be muted somewhat.

Thank you. We have muted our tone when discussing metacognitive issues in SCD throughout the manuscript.

11. Line 54 of page 11: there are many biomarkers that can be used to investigate preclinical AD - particularly with regards to CSF and PET neuroimaging. And APOE genotyping is often done in studies - it is unclear what the authors are referring to here?

Thank you and we agree that this part was unclear. We had meant that in clinical practice testing for these biomarkers is not widely available and we have removed the comment.

Reviewer: 2

Reviewer Name: Sietske Sikkes

Institution and Country: Alzheimer Center, VU University Medical Center, the Netherlands

Competing Interests: None declared

This is a timely and interesting systematic review focusing on interventions for SCD, which is a target population of growing interest, because of their potential higher risk of developing dementia. Despite the relevance for this target populations, I do have some concerns regarding the methodology and the conclusions drawn by the authors.

1. First of all, the focus of the paper is unclear: is this aimed at SCD in the context of AD or is it focused on SCD as a 'functional memory disorder'? Both terms are used in the manuscript, and have very different implications for which studies should be included in the review and the literature that should be discussed. Related to that, it is unclear why the authors chose to combine all potential interventions into one review, for complementary medicine it is noted that 'this has not been evaluated before', which does not seem a legitimate reason to fit it in the current study. From the introduction, it is not clear what the current study is adding to previous systematic reviews for this population.

Thank you for the comment. The aim of the review is to systematically review the literature for SCD in a manner that will be useful for clinicians. We argue that realistically in the clinic it is difficult to discern between SCD in the context of AD and non-preclinical AD because investigations for biomarkers are not readily available, and the diagnosis of SCD is made according to clinical judgement.

We have conducted a broad review because there is currently no evidence based consensus on best management for this patient group. That is why complementary medicine is also included in the review.

This current study does add to previous systematic reviews. Firstly, there have been new relevant

RCTs since the last published review (Smart et al., 2017). Furthermore, we are looking at psychological well-being and metacognition as outcomes and we have justified why these are important in the introduction in addition to objective cognitive performance outcomes used in previous systematic reviews.

2. The literature included in this paper is insufficient: SCD in the context of AD is missing some highly relevant citations on the definition and operationalization of SCD that resulted from the SCD working group (e.g. Molinuevo et al. 2017); as well as literature regarding the potential to prevent AD in SCD. Also regarding the concept of 'metacognition' which is mentioned as a potential cause for SCD the literature is not adequately discussed; this could indeed be a cause, but the authors do not discuss relevant literature indicating that SCD is preceded by actual cognitive decline (e.g. Koppa et al. 2015).

Thank you we have included the references above in the introduction, highlighting that SCD may be preceded by subtle cognitive decline as follows 'patients presenting with SCD may already have subclinical and subtle cognitive decline suggesting an early manifestation of neurodegeneration (Koppa et al., 2015).

We have also revised the 6th paragraph of the introduction (lines 78-85) to highlight the relevance of metacognition:

Given the lack of evidence-based treatments to offer patients with SCD in the clinic, alternative options need to be sought. Metacognition, an individual's insight into their own cognitive functioning, may be of relevance as a therapeutic target because, by definition, SCD involves an individual's underestimation of their cognitive ability relative to objective testing. Metternich²¹ demonstrated that this patient group had significantly poorer metacognitive abilities than controls. As an over-estimate of cognitive impairment may be related to difficulties in evaluating cognitive performance (i.e. metacognitive difficulties), improving metacognitive ability may reduce the subjective cognitive impairment experienced by these patients.

3. An important issue lacking in the discussion is the definition of SCD, which may differ by differences in populations (e.g. memory clinic versus population-based), but also by means of the high variability in measurement instruments that are used to define SCD (Rabin et al. 2015).

Thank you we agree and have added this to the first paragraph of the limitations in the discussion section.

4. The authors chose a very broad approach, looking at all potential interventions in all possible SCD populations (population-based vs memory-clinic based). This is comparing apples to oranges, and severely limits the usefulness of this paper. The authors should incorporate these limitations into the discussion.

We have discussed the heterogeneity of the interventions and lack of memory-clinic based studies as limitations in the final two paragraphs of the limitations section in the discussion.

5. The authors should be more cautious in their conclusions: they note on the one hand that there is a lack of high-quality evidence (RCT's) for interventions on SCD - but on the other hand they conclude that several interventions have no effect. If there's insufficient evidence, this cannot be concluded based on this study.

In our conclusions we have changed the word evidence to research to highlight that there is inadequate research for many of these interventions to prevent any misunderstanding that we were

trying to say that we can robustly conclude that there is insufficient effect of the interventions.

6. The authors state that they included all studies with SCD as a target population "although most studies used other nomenclature" - what nomenclature was used, and how did the authors define SCD?

Under inclusion and exclusion criteria we have written 'SCD which we defined as being self-reported cognitive complaints, without objective evidence of deficits on cognitive testing and unaccounted for by medical or psychiatric causes' (Jessen et al., 2014) to explain how we defined SCD.

7. Important details regarding the methods are not available: The majority of the methods section (e.g. quality assessment, creating composite scores) is in the appendix, not in the main paper - making it difficult to interpret the quality of this systematic review. In addition, how did the authors decide an intervention was 'comparable'? How did the authors decide on the target population? This information is missing from the paper.

Thank you, we have moved this information to the methods section from the supplementary material in order to clarify the methods. We have classified the target populations as follows 'SCD defined as SCD which we defined as being self-reported cognitive complaints, without objective evidence of deficits on cognitive testing and unaccounted for by medical or psychiatric causes'.

8. The authors a priori put much weight on a 'global cognitive measure' - why is that? Would the aim not be to reduce SCD?

We only put weight on global cognition for studies which investigated cognitive interventions. This is because the literature suggests cognitive interventions often improve the domain being trained but the clinical utility of this is limited (Huntley et al., 2015). Global cognition improvement may therefore be more important.

9. The references to functional memory disorder do not seem to relate to the concept of SCD.

Thank you. We have removed references to FMD as we agree that it is confusing to group with SCD and may be subtly different.

10. The authors highlight important differences between memory-clinic SCD and population-based SCD in the introduction; so grouping interventions over those populations does not seem useful. This should be part of the discussion.

Thank you we agree and have included this in the discussion under the 5th paragraph of the limitations:

The goal of research into interventions must be for them to be translational into clinical practice. Unfortunately, only one study³⁰ included in the meta-analyses recruited patients from a healthcare setting whilst the remainder relied on convenience sampling of volunteers with subjective cognitive complaints from the community. This may limit how translatable our findings are to clinical populations.

Reviewer: 3

Reviewer Name: Andrea Tales

Institution and Country: Swansea University, Wales, UK Competing Interests: None except I am a co-author on a previous systematic review [Smart et al 2017] cited by the authors of this systematic review.

Please can the authors consider the following points.

On page 2, in the conclusions section of the abstract the authors state that 'group psychological interventions improve psychological well-being and are likely to improve metacognition' but do the results warrant such a statement. My concern is that the analysis outcome is not robust enough, in terms of the number and quality of the studies, to claim that group psychological interventions are likely to improve metacognition ...how is 'likely' defined?

Thank you, we have changed the wording to 'may' which we agree would be more appropriate given the number and quality of studies.

Page 2: Under strengths and limitations of this study: What do the authors mean by the statement that interventions on psychological well-being and metacognition as it a more naturalistic outcome? What does naturalistic mean?

By naturalistic we meant an outcome measure which would have relevance to everyday life rather than objective cognitive performance where subtle improvements are unlikely to improve everyday functioning. We have deleted the sentence to prevent confusion as suggested by the reviewer.

Page 3. In terms of SCD prevalence, some of these references [4-7] are quite old. Is there newer data on this; especially as interest in SCD has increased since the publication of these papers?

There was no newer data that could be found.

Can the use of the word 'sufferers' (page 3) be avoided.

It has been removed.

I'm a little confused by your description of SCD in the introduction on page 3. Do all people with SCD have increased amyloid deposition in their brains? What about for those for whom SCD does not represent prodromal AD or other dementia?

Apologies that this was unclear. At the end of the second paragraph in the introduction, we have clarified that the brain changes are likely to arise in preclinical AD but the studies which have shown this did not have any way of distinguishing between preclinical AD SCD and non-preclinical AD SCD beforehand

The section on metacognition in the introduction is rather confusing; especially the second sentence and not sure of the context with respect to the present systematic review...more detail and context would be appropriate here.

We have tried to make this paragraph, the 6th in the introduction of revised manuscript, clearer and highlighted that metacognitive deficits are more likely of relevance to non-preclinical AD SCD patients.

In the section on purpose of current review...again the term 'naturalistic' is used but without a detailed description of what this means and its context. I would also question the statement that deficits in metacognition may underpin SCD...is SCD as specific as this...More information is needed as to why these outcomes were chosen, i.e., biological/behavioural basis, and are these likely to occur in isolation? What about the fact that some SCD may be caused by medical issues or dementia and how does this fact relate to the outcome choices made for this review? Could this be linked to the decision to also include pharmacological interventions.

Through our introduction and especially in the last paragraph of the introduction under the purposes of current review we have tried to justify why we have chosen the outcomes we have.

In the methods section the authors state that participants had SCD. How was this judged bearing in mind the issue of nosology in SCD?

This is now clarified in the methods section under inclusion and exclusion criteria.

How many participants in total were included in the meta analysis?

This has been added in the results section.

On page 8 under the 'mindfulness' section please can the authors describe fully why they state that key limitation to Smart et al's study was that the control group received psycho-education....surely this is common practice when comparing different groups and the specificity of an intervention to disease or disorder?

Thank you, we have removed this comment as we agree that psycho-education is a common control intervention.

Discussion

The word 'that' is included twice on line 2 under main findings.

Thank you, this has been corrected.

Paragraph 2; the authors state that, even though the effect size was only moderate, psycho-education is a promising cost effective intervention...The outcome was based only on relatively few studies. Does this evidence warrant such a statement?

Thanks, we agree and have changed the wording and removed 'cost-effective'.

I feel that there is a general bias throughout this paper with respect to the description of the aetiology of SCD in relation to a behavioural/ psychogenic aetiology; the issue of the effect on results of people for whom SCD represents prodromal dementia is not much discussed.

We have tried to focus on the how these findings will help us as clinicians in memory services, cognitive neurology clinics etc.

How, in the papers included in the meta analysis, SCD (or what it was called) was diagnosed is not described for all studies.

We have added this under the inclusion and exclusion criteria in the methods now.

On page 12 under the heading 'Implications' the authors state that this review demonstrates the need for higher quality studies in SCD research with a distinction made between SCD in pre-clinical AD and otherwise' but in what way does the review demonstrate this and how can one determine for whom SCD represents pre-clinical AD?

Determining for whom SCD represents preclinical AD and is through investigating biomarkers but we feel that is not possible routinely in most healthcare settings and that question is beyond the realm of

our review.

The review demonstrates this because based on previous literature we have reviewed there are no studies which have robustly distinguished between these two groups of SCD patients. Therefore, this is a gap in the literature and if filled will help identify targeted treatments for each patient group.

Reviewer: 4

Reviewer Name: Nick Garrett

Institution and Country: AUT, Auckland, New Zealand Competing Interests: None declared

There are two minor issues:

- page 7 lines 41-47 the criteria are relevant but as we are dealing with articles the issues are that a priori power calculations and randomisation are not reported rather than not carried out, suggest minor rewording to demonstrate this.

Thank you, we have revised the Assessment of trial quality section as follows: 'Five criteria assessing quality and bias were assessed based on whether studies reported the following' in the assessment of quality section in the methods section.

- the written results section is ordered by intervention type, whereas the figures referred to are ordered by outcome. It makes it very confusing to follow. Suggest either more text in results section to clarify or to make sure the results and figures correspond.

We appreciate reviewer 4's comments about the written results being ordered by intervention and the forest plots by outcomes. We considered the suggestion to make them both the same. However, if the written section is not ordered by intervention we believe it effects the readability, as currently the interventions can be described together more easily and prevents repeating as would be the case if ordered by outcomes. The forest plot are ordered by outcome, however, we feel this is the best format to visualise the results so that different interventions (grouped together) can be compared to each other for a particular outcome more easily.

VERSION 2 – REVIEW

REVIEWER	Rachel Buckley Massachusetts General Hospital, Boston, MA, United States
REVIEW RETURNED	08-May-2018
GENERAL COMMENTS	None
REVIEWER	Andrea Tales Swansea University, Wales
REVIEW RETURNED	18-May-2018
GENERAL COMMENTS	No additional comments. All my queries and concerns have been addressed.